# The impact of Traditional Chinese Medicine utilization on life expectancy and mortality

**Jui-Yi Wang[1], Hsien-Chang Wu[2,3], Jing-Shiang Hwang[4], Wei-Cheng Lo[1,5]***

**1** School of Public Health, College of Public Health, Taipei Medical University, Taipei, Taiwan, **2** School of Post-Baccalaureate Chinese Medicine, Tzu Chi University, Hualien, Taiwan, **3** Department of Chinese Medicine, Taipei Tzu Chi Hospital, The Buddhist Tzu Chi Medical Foundation, New Taipei City, Taiwan, **4** Institute of Statistical Science, Academia Sinica, Taipei, Taiwan, **5** Master Program in Applied Epidemiology, College of Public Health, Taipei Medical University, Taipei, Taiwan

* nicholaslo@tmu.edu.tw

## Abstract

### Purpose

Traditional Chinese Medicine (TCM) is extensively utilized in Asian societies and has shown potential benefits in improving survival rates of patient with specific diseases and anti-aging effects. However, its impact on life expectancy of general population remains relatively unexplored. This study aimed to investigate the impact of TCM utilization on mortality risk and life expectancy in Taiwan.

### Methods

A nationwide longitudinal cohort study was conducted using data from the Taiwan National Health Interview Survey linked with the National Health Insurance Research Database. Cox proportional hazard models were used to calculate the risk of all-cause mortality between frequent TCM users (≥20% of outpatient visits) and non-frequent users (<20%). A rolling extrapolation algorithm was used to estimate lifetime survival functions, and inverse probability of treatment weighting was integrated to adjust for confounding variables.

### Results

The study included 12,176 participants (1,596 frequent TCM users and 10,580 non-frequent TCM users) aged ≥55 years, with a median follow-up duration of 10.49 years. After adjustment for confounding factors, frequent TCM users had a longer life expectancy compared to non-frequent TCM users, with a difference of 1.37 years (95% CI: 0.22–3.32). Higher TCM utilization was associated with reduced mortality risk (HR: 0.89, 95% CI: 0.80–0.99). Results remained consistent across dose-response analysis and time-dependent exposure models.

**Data availability statement:** The data used in this analysis are not owned by the authors and therefore cannot be shared publicly. To acquire access to the individual-level data, interested researchers must complete an application form, submit a research proposal, and provide documentation of institutional review board approval to the Health and Welfare Data Science Center, Taiwan (https://dep.mohw.gov.tw/DOS/cp-5119-59201-113.html). The Center will review these materials and grant access to eligible researchers who meet the criteria for accessing confidential data. It should be noted that authorized researchers will be granted access to the data under the same conditions and procedures as the authors.

**Funding:** This work was supported by the National Science and Technology Council (Grant number: NSTC112-2314-B-038-073-MY3) in Taiwan. The funders had no role in study design, data collection and analysis, decision to publish, or preparation of the manuscript. There was no additional external funding received for this study.

**Competing interests:** The authors have declared that no competing interests exist.

## Conclusions

This study suggests that higher TCM utilization is associated with longer life expectancy and lower mortality risk among older adults in Taiwan. Further studies are warranted to clarify potential mechanisms and to explore how TCM utilization may complement conventional healthcare in addressing the needs of aging populations.

## Background

Traditional Chinese Medicine (TCM) is widely utilized across Asian societies for both daily health maintenance and the treatment of various medical conditions [1–5]. TCM encompasses practices such as Chinese herbal medicine, acupuncture, and manipulative treatments, and is a cornerstone of complementary and alternative medicine therapies. In countries like Taiwan, Japan, Korea, and China, TCM has been institutionalized to a degree nearly equivalent to that of conventional Western medicine [6,7]. Practitioners are officially certified and regulated, and expenses related to TCM therapies are covered by national health insurance schemes [8–10].

As an integrative treatment approach, TCM has been suggested to provide potential benefits in improving patient survival rates and anti-aging effects. Multiple observational studies have found that combining standard Western medical treatments, such as surgery and chemotherapy, with adjunctive TCM therapies associated with reduced mortality risk among patients with advanced breast, gastric, lung, and liver cancers [11–16]. Additionally, recent studies have suggested that Chinese herbal medicines or their active components may delay aging and prevent age-related diseases [17,18]. Polysaccharides, monopolysaccharides, and sesquiterpenes in TCM have been reported to exhibit anti-inflammatory, antitumor, antiviral, and sedative effects, making them potential sources for the development of anti-aging drugs [19,20].

Nevertheless, the impact of TCM utilization on life expectancy (LE) of general population has been relatively underexplored. One study has evaluated the Chinese healthcare system reform utilized ecological level data to investigate the impacts of TCM on population health outcomes and medical expenditures [21]. The analysis revealed that an increase of one TCM physician per 100,000 population was associated with a reduction of 1.944 excess deaths, an increase of 5.84 days in male LE, and a decrease of 0.051% in average medical expenditures among both urban and rural residents. However, the use of the number of TCM physicians as a proxy for TCM exposure may introduce bias, and the ecological analysis design could result in numerous individual-level confounders not being adequately accounted for.

The principle of TCM treatment is to enable the body to reach a state of normal harmony through drug treatment, emphasizing the strengthening of the body's inherent immunity to cure diseases and providing individualized and precise treatment. The World Health Organization (WHO) has underscored the significance of traditional medicine and has appealed for global policies to support its development, as outlined in the WHO Traditional Medicine Strategy 2014–2023 [22]. Therefore, to quantify the

impact of TCM utilization on overall health and LE will serve as empirical support for promoting the use of TCM and the development of TCM-based drugs. The objective of this study is to investigate the impact of TCM utilization on mortality risk and LE in Taiwan. Using a nationally representative cohort with over 10,000 participants, a rolling extrapolation algorithm was applied to estimate the lifetime survival function and LE of comparative study cohorts of frequent and non-frequent TCM users.

## Methods

### Study design, settings, and population

A nationwide longitudinal cohort study was conducted. We used data from the Taiwan National Health Interview Survey (NHIS) linked with the National Health Insurance Research Database (NHIRD) to determine participants' utilization of TCM services. The study sample comprised individuals who participated in the Taiwan NHIS in 2001, 2005, 2009, and 2013, with response rates of 93.8%, 80.6%, 84.0%, and 75.2%, respectively [23]. The NHIS is a cross-sectional survey that adopted a multistage stratified sampling scheme to obtain a nationally representative sample of the Taiwanese population. Baseline data on participants' sociodemographic and behavioral factors were collected through in-person interviews. Details of the NHIS design and sampling scheme have been previously reported [24]. Participants were classified as frequent TCM users if outpatient visits for TCM services, including Chinese herbal formula, acupuncture, and remedial massage (Tuina), accounted for more than 20% of their total outpatient visits during the 18 months preceding the baseline. Those whose TCM visits constituted 20% or less were classified as non-frequent TCM users. Based on national health insurance data, TCM typically accounts for about 10% of outpatient visits in Taiwan, indicating most individuals use it occasionally [25]. We therefore used a conservative 20% threshold to identify those for whom TCM constitutes a relatively frequent and substantial part of their healthcare use. A total of 20,898 participants aged 55 years and older were initially included. Exclusions were made for participants who refused to link their NHIS data to NHIRD records (n = 3,251), those with a history of cancer (n = 958) or autoimmune rheumatic disease and other rare disease (n = 866) or, those with no healthcare utilization record or fewer than five medical visits in the past year and a half before baseline (n = 1,598), and those with missing covariate data (n = 2,049). The final analytical sample included 12,176 participants at baseline, comprising 1,596 frequent TCM users and 10,580 non-frequent TCM users. Participants were followed until the end of the study period (December 31, 2020) or until death, as confirmed by data from Taiwan's vital registry (Fig 1). The data were accessed for research purposes from 6th August 2023–24th October 2024. This study was approved by the Taipei Medical University-Joint Institutional Review Board (TMU-JIRB- N202304099). All human research was conducted according to the Declaration of Helsinki. Because the database contains only deidentified data, the IRB waived the requirement for written informed consent.

### Covariates

We obtained study participants' sociodemographic characteristics and lifestyle factors from the NHIS database, and their baseline disease history from the NHIRD. The NHIS data collection involved in-person interviews and structured questionnaire survey [24]. We accounted for various potential confounders and covariates, including year of enrollment, baseline age, sex, education level, marital status, monthly household income, employment status, BMI, smoking habits, alcohol consumption, betel nut use, leisure-time physical activity, and adequate intake of fruits and vegetables. Additionally, we included baseline disease history such as cardiovascular disease, diabetes mellitus, chronic lung diseases (asthma or chronic obstructive pulmonary disease), chronic kidney diseases, chronic liver disease, and dementia (S1 Table and S1 File).

### Statistical analysis

Despite a median follow-up duration of 10.49 years, a significant proportion of participants remained alive by the end of follow-up. A rolling extrapolation algorithm can be used to estimate the lifetime survival function of study cohorts [26]. This extrapolation method has been successfully implemented in many studies to estimate the loss or gain of LE in cohorts

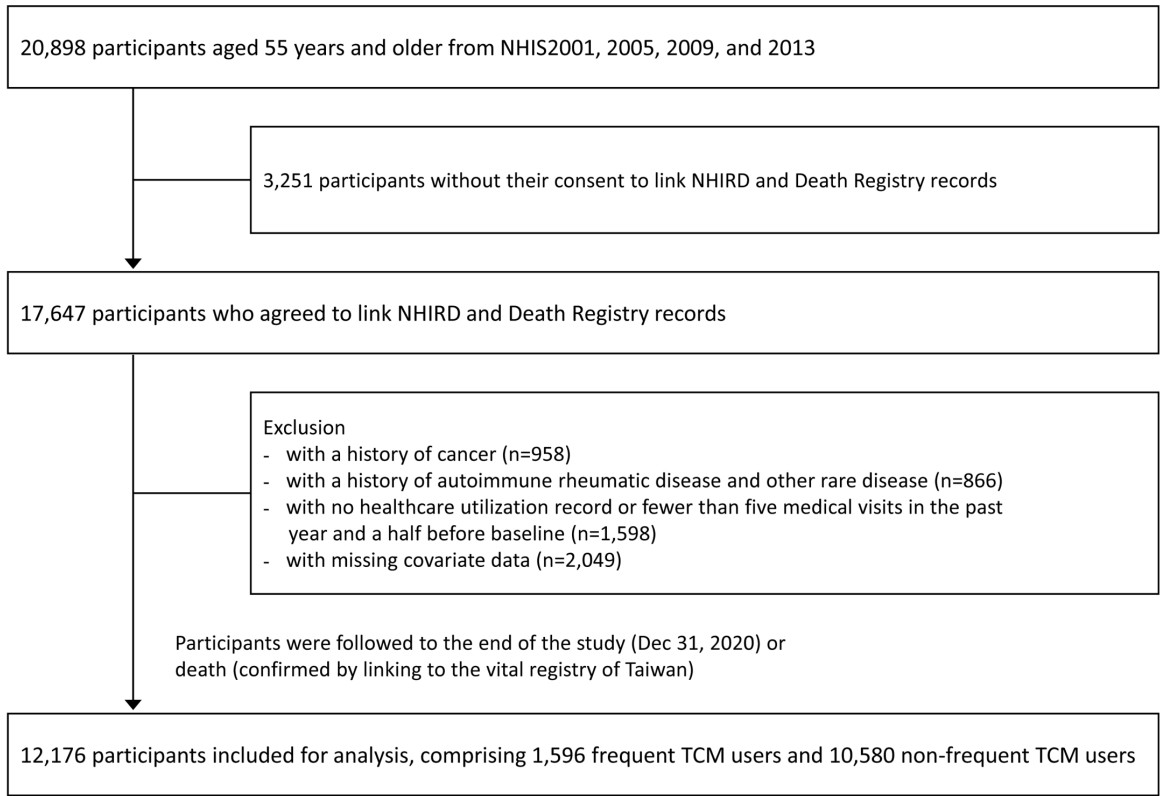

**Fig 1. Flowchart of the study population selection.** NHIS: National Health Interview Survey; NHIRD: National Health Insurance Research Database; TCM: Traditional Chinese Medicine.

with specific health conditions and exposures [27–30]. LE for each study cohort was derived by integrating the corresponding extrapolated lifetime survival function. To make a fair comparison of LE between the comparative cohorts of frequent and non-frequent TCM users, we used an inverse probability of treatment weighting (IPTW) approach to adjust for confounding variables which could influence both utilization of TCM and LE. That is, we first calculated the propensity score for each participant, which is the estimated probability of being in a particular cohort based on a logistic regression model. We then assigned each individual a weight, which is the inverse of their propensity score, to create two weighted cohorts or cohorts of weighted samples. To prevent extreme weights from disproportionately influencing the results, values below the 1st percentile and above the 99th percentile were truncated to the corresponding cutoff values. Next, we applied the rolling extrapolation algorithm to the weighted survival times to obtain lifetime survival functions for the two weighted groups [31,32]. We calculated standardized mean difference (SMD) of the covariates between the two groups of weighted samples to assess balance of covariates in the two groups, with SMD < 0.1 indicating an acceptable balance. The difference in LE between the two weighted cohorts represented the estimated LE difference attributable to differential levels of TCM utilization. All estimates, standard errors, and 95% confidence intervals (CIs) were obtained using the R package iSQoL2, which can be downloaded from here: http://sites.stat.sinica.edu.tw/isqol/.

To assess the differential mortality risk between frequent and non-frequent TCM users, we used Cox proportional hazard models to calculate adjusted hazard ratios (HRs) and corresponding 95% CIs for all-cause mortality associated with TCM utilization. A series of sensitivity analyses were conducted to examine the robustness of results under varying exposure definition. First, we treated TCM utilization as continuous variable, quantified by the percentage of outpatient visits for

TCM services relative to total outpatient visits in the 18 months preceding the baseline. Subsequently, we stratified TCM utilization into four distinct categories based on the proportion of TCM-related outpatient visits: < 10%, 10–20%, 20–30%, and >30%. Moreover, to account for the dynamic nature of TCM utilization, we implemented a time-dependent variable approach. In this time-dependent Cox model, annual TCM utilization was defined as the proportion of TCM outpatient visits relative to total outpatient visits in the preceding year. This variable was updated each year during follow-up to reflect temporal changes in individual patterns of TCM use, allowing a more precise analysis of its association with mortality risk. These models were adjusted for the effects of the aforementioned covariates. Statistical significance was set at a 2-tailed value of P < 0.05.

## Results

### Baseline characteristics

The characteristics of the study participants, a total of 12,176 individuals, including 1,596 frequent TCM users and 10,580 non-frequent TCM users, are shown in Table 1. The mean age for frequent TCM users was 64.1 ± 7.7 years, compared to 67.1 ± 8.9 years for non-frequent TCM users. Females comprised 58.2% of the frequent TCM user group and 49.7% of the non-frequent TCM user group. Frequent TCM users were more likely to be married, possess higher educational attainment, report lower household income levels, and less unemployed. Regarding lifestyle behaviors, frequent TCM users were less likely to engage in smoking, alcohol consumption, and betel nut use, and more inclined to participate in regular leisure-time physical activity and maintain adequate vegetable and fruit intake. The mean BMI was 23.9 ± 3.3 kg/m² for frequent TCM users and 24.5 ± 3.6 kg/m² for non-frequent TCM users. Additionally, the prevalence of cardiovascular disease, diabetes mellitus, and chronic lung diseases, chronic liver disease, and dementia was lower among frequent TCM users. After the IPTW adjustment, the participants' characteristics between the frequent and non-frequent TCM users were well balanced with a SMD < 0.1 (Table 1).

### Impact on life expectancy

Using the rolling extrapolation method, we estimated the lifetime survival curves for the weighted cohorts of frequent and non-frequent TCM user (Fig 2). Based on the extrapolated survival curves from the original samples, we estimated a LE of 21.7 years (95% CI: 19.1–23.4) for frequent TCM users with a mean age of 64.1 years, and 17.6 years (95% CI: 17.0–18.1) for non-frequent TCM users with a mean age of 67.1 years. Crude estimates suggested a slightly longer expected lifespan among frequent TCM users (85.8 vs. 84.7 years). After adjustment for potential confounders using weighted samples, frequent TCM users showed a longer LE compared to non-frequent TCM users, with a difference of 1.37 years (95% CI: 0.22–3.32, p < 0.05) (Table 2). Stratified analyses revealed that among males, frequent TCM users had a 2.39-year higher LE compared to non-frequent TCM users (95% CI: 0.27–4.21, p = 0.02), whereas among females, frequent TCM users had a 1.22-year lower LE, though not statistically significant (95% CI: −3.49–1.29, p = 0.33). When stratifying by smoking status, current or former smokers who were frequent TCM users had a significantly longer LE than non-frequent TCM users (3.31 years, 95% CI: 0.07–6.41, p = 0.04). However, no significant difference was found among never-smokers (LE difference: 0.35 years, 95% CI: −2.32–2.57, p = 0.79). Similar patterns were observed when stratifying by alcohol use. Among drinkers, frequent TCM users showed a longer LE compared to non-frequent TCM users, although this difference did not reach statistical significance (LE difference: 2.81 years, 95% CI: −1.49–5.41, p = 0.13).

### Traditional Chinese medicine utilization and mortality risk

During the follow-up period, a total of 4,023 deaths were recorded. In Cox regression model adjusting for age, sex, enrollment year, education level, marital status, monthly household income, employment status, lifestyle factors, and comorbid diseases, frequent TCM user was associated with a modestly lower risk of all-cause mortality, with a HR of 0.89 (95% CI,

**Table 1. Baseline demographic and lifestyle characteristics of study population before and after the IPTW adjustment.**

| | Before IPTW | | | | | | After IPTW | | | |
| --- | --- | --- | --- | --- | --- | --- | --- | --- | --- | --- |
| | Frequent TCM user | | Non-frequent TCM user | | | | Frequent TCM user | Non-frequent TCM user | | |
| N | 1596 | | 10580 | | | | 12305.1 | 12174.6 | | |
| **Demographic characteristics** | N | (%) | N | (%) | SMD | *P*-value | N (%) | N (%) | SMD | *P*-value |
| Age, Mean (SD) | 64.1 | (7.7) | 67.1 | (8.9) | 0.360 | <0.0001 | 66.7 (8.8) | 66.8 (8.7) | 0.005 | 0.879 |
| Age group | | | | | 0.339 | <0.0001 | | | 0.024 | 0.894 |
| 55–59 yrs | 593 | (37.1) | 2772 | (26.2) | | | 3330.4 (27.1) | 3364.3 (27.6) | | |
| 60–69 yrs | 652 | (40.9) | 3993 | (37.7) | | | 4642.1 (37.7) | 4643.1 (38.1) | | |
| 70–79 yrs | 270 | (16.9) | 2748 | (26.0) | | | 3179.9 (25.8) | 3018.9 (24.8) | | |
| 80+ | 81 | (5.1) | 1067 | (10.1) | | | 1152.7 (9.4) | 1148.2 (9.4) | | |
| Sex | | | | | 0.171 | <0.0001 | | | 0.006 | 0.833 |
| Male | 668 | (41.8) | 5325 | (50.3) | | | 6095.5 (49.5) | 5991.4 (49.2) | | |
| Female | 928 | (58.2) | 5255 | (49.7) | | | 6209.6 (50.5) | 6183.2 (50.8) | | |
| Education | | | | | 0.153 | <0.0001 | | | 0.018 | 0.825 |
| Less than elementary school | 920 | (57.6) | 6884 | (65.1) | | | 7993.0 (65.0) | 7805.4 (64.1) | | |
| High school | 464 | (29.1) | 2571 | (24.3) | | | 2991.5 (24.3) | 3031.7 (24.9) | | |
| College or above | 212 | (13.3) | 1125 | (10.6) | | | 1320.6 (10.7) | 1337.4 (11.0) | | |
| Marriage | | | | | 0.049 | 0.1961 | | | 0.012 | 0.925 |
| Married/cohabiting | 1202 | (75.3) | 7787 | (73.6) | | | 9144.2 (74.3) | 8988.2 (73.8) | | |
| Never married | 34 | (2.1) | 198 | (1.9) | | | 237.3 (1.9) | 231.2 (1.9) | | |
| Others* | 360 | (22.6) | 2595 | (24.5) | | | 2923.5 (23.8) | 2955.1 (24.3) | | |
| Household income# | | | | | 0.138 | <0.0001 | | | 0.017 | 0.856 |
| Low | 390 | (24.4) | 2371 | (22.4) | | | 2747.9 (22.3) | 2762.4 (22.7) | | |
| Median | 689 | (43.2) | 4086 | (38.6) | | | 4767.1 (38.7) | 4771.8 (39.2) | | |
| High | 517 | (32.4) | 4123 | (39.0) | | | 4790.1 (38.9) | 4640.4 (38.1) | | |
| Employment status | | | | | 0.111 | <0.0001 | | | 0.014 | 0.665 |
| Employed (included homemaker & retired) | 1349 | (84.5) | 8495 | (80.3) | | | 9878.3 (80.3) | 9842.5 (80.8) | | |
| Unemployed | 247 | (15.5) | 2085 | (19.7) | | | 2426.7 (19.7) | 2332.1 (19.2) | | |
| **Lifestyle factors** | | | | | | | | | | |
| Cigarette smoking | | | | | 0.188 | <0.0001 | | | 0.017 | 0.882 |
| Never smokers | 1264 | (79.2) | 7525 | (71.1) | | | 8804.6 (71.6) | 8787.2 (72.2) | | |
| Former smokers | 131 | (8.6) | 1274 | (12.1) | | | 1425.1 (11.6) | 1410.0 (11.6) | | |
| Current smokers | 195 | (12.2) | 1781 | (16.8) | | | 2075.3 (16.9) | 1977.3 (16.2) | | |
| Alcohol use | | | | | 0.099 | 0.0113 | | | 0.019 | 0.950 |
| Nonconsumers | 1019 | (63.8) | 6811 | (64.4) | | | 7893.5 (64.1) | 7828.3 (64.3) | | |
| Infrequent consumers | 410 | (25.7) | 2506 | (23.7) | | | 2913.2 (23.7) | 2915.4 (23.9) | | |
| Regular consumers | 156 | (9.8) | 1089 | (10.3) | | | 1285.8 (10.4) | 1245.7 (10.2) | | |
| Excess consumers | 11 | (0.7) | 174 | (1.6) | | | 212.6 (1.7) | 185.1 (1.5) | | |
| Betel nut use | | | | | 0.101 | 0.0023 | | | 0.018 | 0.878 |
| Never user | 1490 | (93.4) | 9619 | (90.9) | | | 11220.0 (91.2) | 11107.8 (91.2) | | |
| Former user | 67 | (4.2) | 533 | (5.0) | | | 642.2 (5.2) | 600.0 (4.9) | | |
| Current user | 39 | (2.4) | 428 | (4.1) | | | 442.8 (3.6) | 466.8 (3.8) | | |
| Leisure-time physical activity | | | | | 0.106 | 0.0001 | | | 0.013 | 0.665 |
| Yes | 991 | (62.1) | 6018 | (56.9) | | | 7001.8 (56.9) | 7008.7 (57.6) | | |
| No | 605 | (37.9) | 4562 | (43.1) | | | 5303.3 (43.1) | 5165.9 (42.4) | | |

*(Continued)*

**Table 1.** (Continued)

| | Before IPTW | | | | | | After IPTW | | | |
| --- | --- | --- | --- | --- | --- | --- | --- | --- | --- | --- |
| | Frequent TCM user | | Non-frequent TCM user | | | | Frequent TCM user | Non-frequent TCM user | | |
| Adequate intake of vegetables | | | | | 0.083 | 0.0031 | | | 0.026 | 0.452 |
| Yes | 1492 | (93.5) | 9657 | (91.3) | | | 11174.3 (90.8) | 11146.9 (91.6) | | |
| No | 104 | (6.5) | 923 | (8.7) | | | 1130.8 (9.2) | 1027.7 (8.4) | | |
| Adequate intake of fruits | | | | | 0.136 | <0.0001 | | | 0.023 | 0.485 |
| Yes | 1200 | (75.2) | 7309 | (69.1) | | | 8471.5 (68.8) | 8509.2 (69.9) | | |
| No | 396 | (24.8) | 3271 | (30.9) | | | 3833.6 (31.2) | 3665.3 (30.1) | | |
| BMI group | | | | | 0.172 | <0.0001 | | | 0.017 | 0.959 |
| Underweight (BMI < 18.5 kg/m$^2$) | 66 | (4.1) | 351 | (3.3) | | | 398.0 (3.2) | 415.7 (3.4) | | |
| Normal weight (18.5 ≤ BMI < 25 kg/m$^2$) | 1011 | (63.4) | 5963 | (56.4) | | | 7130.0 (57.9) | 6975.2 (57.3) | | |
| Overweight (25 ≤ BMI < 30 kg/m$^2$) | 447 | (28.0) | 3538 | (33.4) | | | 3999.5 (32.5) | 3983.8 (32.7) | | |
| Obese (BMI ≥ 30 kg/m$^2$) | 72 | (4.5) | 728 | (6.9) | | | 777.6 (6.3) | 799.8 (6.6) | | |
| **Comorbid diseases** | | | | | | | | | | |
| Cardiovascular disease | 652 | (40.8) | 6226 | (58.9) | 0.366 | <0.0001 | 7029.0 (57.1) | 6877.8 (56.5) | 0.013 | 0.665 |
| Diabetes mellitus | 196 | (12.3) | 2253 | (21.3) | 0.243 | <0.0001 | 2667.4 (21.7) | 2450.6 (20.1) | 0.038 | 0.283 |
| Chronic lung diseases | 88 | (5.5) | 862 | (8.2) | 0.105 | 0.0003 | 889.5 (7.2) | 949.5 (7.8) | 0.022 | 0.521 |
| Chronic kidney diseases | 32 | (2.0) | 257 | (2.4) | 0.029 | 0.2995 | 300.6 (2.4) | 289.1 (2.4) | 0.004 | 0.892 |
| Chronic liver disease | 117 | (7.3) | 958 | (9.1) | 0.063 | 0.0236 | 1206.2 (9.8) | 1075.9 (8.8) | 0.033 | 0.330 |
| Dementia | 16 | (1.0) | 206 | (2.0) | 0.078 | 0.0086 | 280.7 (2.3) | 222.5 (1.8) | 0.032 | 0.412 |

IPTW: inverse probability of treatment weighting; SMD: standardized mean difference.

*Others: widowed, divorced, separated, or serving as a single parent.

#Household income (US dollars per month): low: < $980; median: $980 to $3260; high: > $3260.

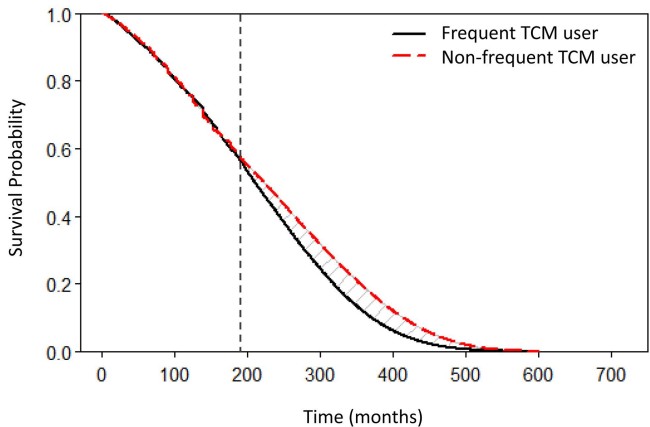

**Fig 2. IPTW-weighted lifetime survival function of frequent and non-frequent TCM user cohorts.** The frequent TCM user group (red dashed line) exhibits longer survival times compared to the non-frequent TCM user group (black solid line). The vertical black dotted line stands for the starting month of extrapolation. TCM: Traditional Chinese Medicine.

**Table 2. Life expectancy and years of life gained of frequent and non-frequent TCM user cohorts.**

| Strata | Frequent TCM user LE (95%CI) | Non-frequent TCM user LE (95%CI) | Years of life gained(95%CI) |
|---|---|---|---|
| Overall | | | |
| IPTW adjustment | 19.25 (17.96, 21.07) | 17.88 (17.25, 18.42) | 1.37 (0.22, 3.32) |
| Sex | | | |
| Male | 18.87 (16.90, 20.82) | 16.48 (15.74, 17.18) | 2.39 (0.27, 4.21) |
| Female | 18.32 (16.21, 20.40) | 19.54 (18.41, 20.55) | −1.22 (−3.49, 1.29) |
| BMI group | | | |
| BMI < 25 kg/m$^2$ | 18.92 (16.92, 20.90) | 17.77 (17.02, 18.48) | 1.15 (−1.00, 3.55) |
| BMI >= 25 kg/m$^2$ | 19.91 (16.70, 22.91) | 18.00 (17.11, 19.20) | 1.91 (−1.47, 4.52) |
| Cigarette smoking | | | |
| Never smokers | 19.39 (17.13, 21.59) | 19.04 (18.19, 19.93) | 0.35 (−2.32, 2.57) |
| Current or former smokers | 18.48 (15.20, 21.62) | 15.17 (14.56, 15.79) | 3.31 (0.07, 6.41) |
| Alcohol use | | | |
| Nonconsumers | 18.42 (16.53, 20.58) | 17.19 (16.56, 18.17) | 1.23 (−0.81, 3.82) |
| Drinkers* | 22.34 (18.12, 24.33) | 19.53 (18.08, 20.50) | 2.81 (−1.49. 5.41) |
| Leisure-time physical activity | | | |
| Yes | 20.62 (18.28, 22.36) | 18.84 (17.93, 19.63) | 1.78 (−0.43, 3.78) |
| No | 18.12 (15.33, 20.64) | 16.71 (15.92, 17.59) | 1.41 (−1.50, 3.83) |
| Adequate intake of vegetables and fruits# | | | |
| Yes | 20.61 (18.32, 22.43) | 19.43 (18.45, 20.25) | 1.18 (−1.20, 3.40) |
| No | 16.31 (13.38, 18.14) | 15.54 (14.77, 16.65) | 0.77 (−2.12, 2.65) |

LE: Life expectancy; IPTW: inverse probability of treatment weighting

*Including infrequent, regular, and excess consumers.

#The intake of both vegetables and fruits is adequate.

0.80–0.99) (Table 3). Dose-response analyses suggested a trend toward reduced mortality risk with higher TCM utilization, though not all categories reached statistical significance. After covariate adjustment, compared to TCM utilization of less than 10%, the HRs for TCM utilization of 10–20%, 20–30%, and greater than 30% were 0.89 (95% CI, 0.79–1.01), 1.01 (95% CI, 0.86–1.12), and 0.81 (95% CI, 0.71–0.93), respectively.

When TCM utilization was treated as a continuous variable using two distinct exposure time windows, either the baseline window (18 months preceding the baseline) or as time-dependent (updated TCM exposure annually). For every 10% increase in TCM utilization at baseline, the HR for all-cause mortality was 0.98 (95% CI, 0.95–1.01). Consistent associations were observed in the time-dependent analysis, with an HR of 0.95 (95% CI, 0.94–0.96) per 10% increase in TCM utilization (Table 3). Additional analyses examining incidence of major chronic diseases yielded results generally consistent with the observed associations for mortality risk (S2 and S3 Tables).

## Discussion

To the best of our knowledge, this is the first study to evaluate the association of TCM utilization with LE and mortality risk in the general population using individual-level empirical data. Our findings suggest that frequent TCM use was associated with modestly increased LE and reduced mortality risk, with a difference of 1.37 years in LE and a HR of 0.89 (95% CI, 0.80–0.99) compared to non-frequent TCM users. These results were generally consistent across various analytical approaches, including binary and multi-categorical exposure classification, dose-response analysis, and time-dependent exposure models, thus strengthening the robustness of our findings.

**Table 3. Hazard Ratios (HRs) and 95% CIs for associations between TCM utilization and all-cause mortality risk.**

| TCM utilization | Deaths | Person-years | Model 1[a]<br>HR (95%CI) | Model 2[b]<br>HR (95%CI) |
|---|---|---|---|---|
| Frequent TCM user vs. non-frequent user | | | | |
| Non-frequent TCM user | 3651 | 109201.7 | 1.00 | 1.00 |
| Frequent TCM user | 372 | 17425.7 | 0.80 (0.72, 0.89) | 0.89 (0.80, 0.99) |
| % of TCM utilization (categorical) | | | | |
| <10% | 3383 | 98222.9 | 1.00 | 1.00 |
| 10-20% | 268 | 10978.8 | 0.86 (0.76, 0.97) | 0.89 (0.79, 1.01) |
| 20-30% | 151 | 5846.7 | 0.96 (0.81, 1.12) | 1.01 (0.86, 1.12) |
| ≥30% | 221 | 11579.0 | 0.71 (0.62, 0.81) | 0.81 (0.71, 0.93) |
| % of TCM utilization (continuous) | | | | |
| per 10% of TCM utilization increase (baseline) | 4023 | 126627.4 | 0.94 (0.92, 0.97) | 0.98 (0.95, 1.01) |
| per 10% of TCM utilization increase (time-varying) | 4023 | 126627.4 | 0.93 (0.92, 0.94) | 0.95 (0.94, 0.96) |

[a]Model 1 adjusted for age, sex, enrolment year, education level, marital status, monthly household income, and employment status.

[b]Model 2 adjusted for age, sex, enrolment year, education level, marital status, monthly household income, employment status, lifestyle factors, and medical history of cardiovascular disease, diabetes mellitus, chronic lung diseases, chronic kidney diseases, chronic liver disease, and dementia.

The observed association aligns with previous research, indicating potential benefits of TCM for specific health conditions and its anti-aging properties. A longitudinal study conducted by Li et al. on 1,988 patients with advanced lung adenocarcinoma reported that subjects utilizing Chinese herbal medicine as an adjuvant to tyrosine kinase inhibitor (TKI) treatment for ≥180 days exhibited significantly reduced mortality rates and improved overall and progression-free survival [33]. Additionally, a large-scale retrospective study in Taiwan, encompassing 79,335 breast cancer patients, suggested a dose-dependent protective effect associated with prolonged use of Danshen (Salvia miltiorrhiza) [34]. Kuo et al. performed a comprehensive analysis of 582,799 adult cancer patients in Taiwan, revealing significantly lower mortality risk among TCM users compared to non-users [35]. Moreover, a real-world study focusing on liver cancer treatment identified Jiawei Xiaoyao San and Chaihu Shugan Decoction as the most efficacious Chinese herbal formulas for improving overall survival [36]. On the other hand, preclinical studies have also elucidated TCM's considerable anti-aging potential, primarily through its anti-inflammatory actions, enhancement of intestinal health, and maintenance of telomere integrity. TCM has been suggested to help address age-related pathologies, including diabetes, neurodegenerative disorders, cardiovascular diseases, and various malignancies, by modulating multiple biological pathways [37]. While these findings are suggestive, the precise mechanisms underlying these effects and the specific impacts of various TCM interventions necessitate further rigorous investigation.

TCM offers a unique advantage in promoting health and treating diseases through its deeply personalized approach, which could potentially contribute to more favorable health outcomes than standardized treatments. Rooted in a holistic perspective, TCM views the human body as a dynamic system seeking Yin-Yang balance [38]. At its core, TCM employs the concept of "Treatment of Differential Syndrome" (TDS), recognizing each patient as a unique individual with specific imbalances and needs [39]. Unlike the often-standardized treatments in Western medicine, TCM practitioners meticulously analyze symptoms and signs to determine a patient's syndrome, considering not only physical manifestations but also genetic, physiological, psychological, spiritual, and social dimensions. This comprehensive assessment enables highly tailored treatments that address root causes rather than merely alleviating symptoms. TCM's combinatorial formulae, synergistically blending multiple herbs and components, are customized based on the patient's specific syndrome and physique characteristics. This personalized approach extends beyond treating diseases to include early intervention and management of sub-healthy states, emphasizing prevention and overall well-being [40]. By focusing on restoring balance

to the entire body system and adapting treatments to the individual's changing needs over time, TCM embodies many principles now emphasized in modern precision and personalized medicine.

In this study, we used the frequency of TCM utilization as a proxy indicator for exposure to TCM practices. This approach not only represents the degree of engagement with TCM but also suggests that higher-frequency TCM users may be more likely to embrace the principles of TCM in disease treatment and health promotion. A knowledge, attitude, and practice survey conducted in Shanghai, focusing on the elderly population, corroborated this notion by demonstrating a strong consistency between cognitive understanding and behavioral practices in the utilization of traditional medicine [41]. TCM users were more likely to possess a heightened health consciousness and pursue a more health-oriented life-style, as presented by the differences observed in baseline characteristics between frequent and non-frequent TCM users in our study. Given the rapid demographic transition and the growing prevalence of multimorbidity, understanding how holistic care models such as TCM can complement conventional medicine is increasingly important. The TCM perspec-tive, which emphasizes mind-body balance and integrative health, offers a useful framework for addressing the complex interplay among chronic diseases and their cumulative effects on quality of life in older adults [38,42]. This multidimen-sional view may help advance a more integrated and person-centered approach to managing multimorbidity. In Asia, the belief in and use of traditional medicine are deeply rooted in cultural and spiritual foundations [43]. However, despite its long-standing history, the empirical evidence base for TCM's population-level effects remains limited [44]. Through the use of real-world longitudinal data and a rigorous research design, this study provides insights into the relationship between TCM utilization and overall health outcomes, providing a reference for future research and the potential role of traditional medicine within modern healthcare systems.

## Strengths and limitations

Our study's strengths lie in its methodological rigor and comprehensive approach. We utilized a large, nationally represen-tative sample with prospective design and near-complete follow-up. The integration of survey data with National Health Insurance claims and vital registry records provided an opportunity to investigate the relationship between TCM utilization on LE in Taiwan. The extrapolation of lifetime survival functions, combined with IPTW adjustment, enhanced the robust-ness and validity of our findings. Several limitations should be acknowledged. First, all analyzed covariates were assessed only at baseline; therefore, changes in these factors over time could not be accounted for, as no repeated measurement data were available. Second, although the models were adjusted for a wide range of demographic, socioeconomic, lifestyle, and health-related variables, the possibility of residual or unmeasured confounding effects cannot be ruled out. TCM users may differ from non-users in health awareness, social support, or health-seeking behaviors that were not fully measured. In addition, regular contact with TCM practitioners may provide non-specific benefits, such as health advice or stress reduction, which are independent of the specific pharmacological or procedural effects of TCM. Third, this study did not differentiate in detail between various types of TCM utilization, such as Chinese herbal formula, acupuncture, and remedial massage (Tuina); therefore, the observed associations should be interpreted as reflecting overall TCM service utilization rather than the effects of specific treatment types. We also acknowledge that the available data do not allow us to determine whether the 20% threshold used to define frequent TCM use represents an optimal cutoff across different populations or health conditions, nor whether it holds equivalent meaning across various TCM modalities. Further stud-ies using more detailed information on TCM use may better define treatment patterns and identify suitable thresholds for frequent use. Fourth, the dose–response analysis assumed a linear association between the proportion of TCM use and mortality risk. Although this approach facilitated interpretation, potential nonlinear patterns such as threshold or plateau effects were not assessed. Future studies applying flexible models, such as restricted cubic splines, may help clarify the functional form of this relationship. Lastly, while healthcare services in Taiwan, including TCM, have a high coverage rate, this study could not evaluate whether participants engaged in other forms of traditional or folk medicine, which may have influenced the outcomes.

## Conclusions

Our study observed an association between TCM use and longer life expectancy, along with a lower risk of all-cause mortality. These associations were consistent across multiple analytical approaches, supporting the robustness of the findings. However, causal inference cannot be established due to the observational study design. The observed associations may reflect not only the effects of TCM utilization but also characteristics of TCM users, the healthcare context, and possible holistic benefits associated with regular engagement with TCM. Our findings provide population-level insights that can inform future studies on TCM's potential role in promoting healthy aging, improving quality of life, and supporting integrated healthcare approaches. While further research is needed to clarify the underlying mechanisms, refine exposure definitions, and account for unmeasured behavioral or psychosocial factors, this study offers a foundation for understanding the potential contribution of TCM as a complementary component of comprehensive healthcare strategies.

## Supporting information

**S1 File. Supplementary methods.**
(PDF)

**S1 Table. ICD codes for baseline comorbidities.**
(PDF)

**S2 Table. Hazard Ratios (HRs) and 95% CIs for associations between TCM utilization (frequent user vs. non-frequent user) and risk of major chronic diseases development.**
(PDF)

**S3 Table. Hazard Ratios (HRs) and 95% CIs for associations between TCM utilization (per 10% of TCM utilization increase) and risk of major chronic diseases development.**
(PDF)

**S1 Data. STROBE checklist cohort.**
(DOC)

## Acknowledgments

We thank Ms. Tsuey-Hwa Hu from the Institute of Statistical Science at Academia Sinica for her professional statistical recommendations and support on this work.

## Author contributions

**Conceptualization:** Jing-Shiang Hwang, Wei-Cheng Lo.

**Data curation:** Jui-Yi Wang, Wei-Cheng Lo.

**Formal analysis:** Jui-Yi Wang, Wei-Cheng Lo.

**Funding acquisition:** Wei-Cheng Lo.

**Supervision:** Jing-Shiang Hwang, Wei-Cheng Lo.

**Visualization:** Jui-Yi Wang, Wei-Cheng Lo.

**Writing – original draft:** Jui-Yi Wang, Wei-Cheng Lo.

**Writing – review & editing:** Hsien-Chang Wu, Jing-Shiang Hwang, Wei-Cheng Lo.

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
