## [Decision Letter · Decision Letter 0]

18 Sep 2025

Dear Dr. Lo,

Thank you for submitting your manuscript to PLOS ONE. After careful consideration, we feel that it has merit but does not fully meet PLOS ONE’s publication criteria as it currently stands. Therefore, we invite you to submit a revised version of the manuscript that addresses the points raised during the review process.

**Thank you for submitting the following manuscript to PLOS ONE. Please revise the manuscript according to the reviewers' comments and upload the revised file.**

We look forward to receiving your revised manuscript.

Kind regards,

Yung-Hsiang Chen, Ph.D.

Academic Editor

PLOS ONE

Journal Requirements:

“This work was supported by the National Science and Technology Council (Grant number: NSTC112-2314-B-038-073-MY3) in Taiwan. The funders had no role in study design, data collection and analysis, decision to publish, or preparation of the manuscript.”

“The authors have no conflict of interest to report.”

6. Please remove all personal information, ensure that the data shared are in accordance with participant consent, and re-upload a fully anonymized data set.

Additional guidance on preparing raw data for publication can be found in our Data Policy (https://journals.plos.org/plosone/s/data-availability#loc-human-research-participant-data-and-other-sensitive-data ) and in the following article: http://www.bmj.com/content/340/bmj.c181.long.

Additional Editor Comments:

Thank you for submitting the following manuscript to PLOS ONE.

Please revise the manuscript according to the reviewers' comments and upload the revised file.

Reviewers' comments:

Reviewer's Responses to Questions

**Comments to the Author**

1. Is the manuscript technically sound, and do the data support the conclusions?

Reviewer #1: Partly

Reviewer #2: Yes

Reviewer #3: Yes

2. Has the statistical analysis been performed appropriately and rigorously?

Reviewer #1: No

Reviewer #2: Yes

Reviewer #3: Yes

3. Have the authors made all data underlying the findings in their manuscript fully available?

Reviewer #1: No

Reviewer #2: Yes

Reviewer #3: Yes

4. Is the manuscript presented in an intelligible fashion and written in standard English?

Reviewer #1: Yes

Reviewer #2: Yes

Reviewer #3: Yes

Reviewer #1: While the study addresses an important public health question regarding Traditional Chinese Medicine (TCM) utilization and its association with mortality and life expectancy, I have major concerns regarding the conceptualization and measurement of the exposure variable, which limit the validity of causal inference.

In the current manuscript, TCM encompasses diverse modalities including herbal medicine, acupuncture, and Tuina (therapeutic massage), which differ substantially in mechanism, intended use, and potential biological impact. Pooling such heterogeneous practices into a single binary or percentage-based measure assumes homogeneity of effect, which is unlikely to be true. This aggregation obscures which component, if any, is responsible for the observed associations, and risks diluting or misattributing effects.

The use of “>20% of outpatient visits in the prior 18 months” as the primary threshold for defining TCM users is arbitrary and lacks empirical justification. This measure does not directly reflect treatment intensity, dose, duration, or quality of TCM received. For example, a patient receiving a few acupuncture sessions could be classified similarly to one on continuous herbal therapy, despite potentially very different physiological effects.

The observed associations may reflect characteristics of individuals who choose TCM, rather than the effects of TCM interventions themselves. TCM users in the cohort were generally younger, healthier, and had more favorable lifestyle factors at baseline. Although IPTW adjustment was employed, unmeasured confounding from health consciousness, diet quality beyond fruit/vegetable intake, social support, and other behavioral factors remains plausible. Regular contact with TCM practitioners may also lead to ancillary benefits (e.g., health advice, stress reduction) unrelated to specific pharmacologic or procedural effects.

Given these limitations, the results should be interpreted cautiously. The study as designed cannot disentangle whether the life expectancy gain is attributable to TCM-specific therapeutic mechanisms or to correlated health behaviors and socio-cultural factors. Current phrasing in the Discussion leans toward causal interpretation, which may overstate the evidence. It is critical to frame the findings as associations and to emphasize the potential role of residual and unmeasured confounding.

Recommendation:

To strengthen the manuscript, I suggest:

- Conducting modality-specific analyses (herbal medicine, acupuncture, Tuina) if feasible.

- Performing sensitivity analyses using alternative TCM exposure thresholds (e.g., 10%, 30%).

- Adjusting for additional markers of health consciousness and healthcare engagement, or employing negative control outcomes to test for unmeasured confounding.

- Revising the Discussion to clearly differentiate between “association” and “causation” and to acknowledge that the observed benefits may reflect characteristics of TCM users rather than TCM per se.

Reviewer #2: 本研究利用台湾全国健康调查和保险数据库探讨中医药使用对老年人预期寿命和死亡风险的影响。研究设计严谨样本量大统计方法先进。研究结果对公共卫生和临床实践具有重要意义。建议稿件经过细微修改后被接受。 使用来自 NHIS 和 NHIRD 的链接数据提供了国家代表性和较长的随访期。应用滚动外推算法来估计终生生存函数结合治疗加权的逆概率 �IPTW� 来控制混杂因素展示了方法论的复杂性。中医使用的定义是“过去 18 个月中医门诊就诊占门诊总就诊量的 20% 以上”是一个关键的暴露标准。它将加强手稿提供简短的理由或引用支持这一阈值的相关文献无论是临床还是方法学。

<Translation performed by Google translate: "This study used the Taiwan National Health Survey and Insurance Database to investigate the impact of TCM use on life expectancy and mortality risk among older adults. The study was rigorously designed, with a large sample size and advanced statistical methods. The findings have important implications for public health and clinical practice. The manuscript was accepted after minor revisions. The use of linked data from the NHIS and NHIRD provided national representativeness and a long follow-up period. The application of a rolling extrapolation algorithm to estimate lifetime survival functions, combined with inverse probability of treatment weighting (IPTW) to control for confounding factors, demonstrates the methodological complexity. The definition of TCM use as "TCM outpatient visits accounting for ≥ 20% of total outpatient visits in the previous 18 months" is a key exposure criterion. Manuscripts would be encouraged to provide brief justification or cite relevant literature supporting this threshold, both clinically and methodologically.">

Reviewer #3: 1.The study employs IPTW, but it is unclear whether extreme weights were addressed. Such weights can compromise stability and validity. Please clarify whether weight stabilization or truncation was applied; if not, provide a rationale.

2.On page 12 of the manuscript, the paragraph beginning with “When analyzing by smoking status, current or former smokers who used TCM had a significantly…” appears to be duplicated. Please revise.

3.This study reports a dose–response relationship between TCM utilization and mortality, expressed as the hazard ratio per 10% increase in TCM use. However, it is not clear whether the analysis assumed a linear relationship. Could the authors clarify if any assessment of non-linearity was performed, for example, using restricted cubic spline models or other approaches? This would help determine whether the association is truly linear or if there are potential threshold or plateau effects.

4.I appreciate the authors’ effort to account for the dynamic nature of TCM utilization using a time-dependent approach, which is appropriate and strengthens the analysis. However, I recommend that the authors provide more methodological details on how the annual proportion of TCM services relative to total outpatient visits was defined and incorporated into the model. Since an individual’s frequency of TCM visits may fluctuate from year to year (e.g., higher in one year and lower in another), it would be helpful to clarify how such variability was handled in the analysis and how the time-varying exposure was formally defined. Clearer reporting of this step would enhance the transparency and reproducibility of the study.

5.The authors may consider conducting additional cause-specific mortality analyses, which could provide further insights into whether the observed association between TCM utilization and overall mortality is attributable to specific causes of death.

**Do you want your identity to be public for this peer review?** For information about this choice, including consent withdrawal, please see our Privacy Policy

Reviewer #1: No

Reviewer #2: No

Reviewer #3: No

---

## [Author Response · Author response to Decision Letter 1]

30 Oct 2025

Dear Editors:

We would like to thank you for your interest in our manuscript entitled “The Impact of Traditional Chinese Medicine Utilization on Life Expectancy and Mortality,” and for inviting us to submit a revised version.

We sincerely appreciate the reviewers’ thoughtful and constructive comments. While we recognize the importance of many of their suggestions, due to the expiration of our approved data access period and constraints in both the scope and format of the available data, we were unable to conduct additional analyses to fully address some of the reviewers’ requests. (The analyses examining the association between TCM use and the incidence of major chronic diseases had been completed previously and were available for reference in this revision.) Nevertheless, we have carefully revised the manuscript to incorporate relevant clarifications and supplementary explanations. In particular, we have refined the discussion and conclusion sections to more appropriately characterize the observed associations and to avoid overstating any causal interpretation.

All modifications in the main text are marked in Track Changes. A detailed, point-by-point response to the reviewers’ comments is also provided. We once again thank the editors and reviewers for their valuable time and insightful feedback, which have greatly improved the quality and clarity of our manuscript.

Thank you for considering our revised submission to PLOS ONE.

Yours sincerely,

Wei-Cheng Lo, PhD, on behalf of all coauthors

Review Comments to the Author

Reviewer #1:

While the study addresses an important public health question regarding Traditional Chinese Medicine (TCM) utilization and its association with mortality and life expectancy, I have major concerns regarding the conceptualization and measurement of the exposure variable, which limit the validity of causal inference.

In the current manuscript, TCM encompasses diverse modalities including herbal medicine, acupuncture, and Tuina (therapeutic massage), which differ substantially in mechanism, intended use, and potential biological impact. Pooling such heterogeneous practices into a single binary or percentage-based measure assumes homogeneity of effect, which is unlikely to be true. This aggregation obscures which component, if any, is responsible for the observed associations, and risks diluting or misattributing effects.

The use of “>20% of outpatient visits in the prior 18 months” as the primary threshold for defining TCM users is arbitrary and lacks empirical justification. This measure does not directly reflect treatment intensity, dose, duration, or quality of TCM received. For example, a patient receiving a few acupuncture sessions could be classified similarly to one on continuous herbal therapy, despite potentially very different physiological effects.

The observed associations may reflect characteristics of individuals who choose TCM, rather than the effects of TCM interventions themselves. TCM users in the cohort were generally younger, healthier, and had more favorable lifestyle factors at baseline. Although IPTW adjustment was employed, unmeasured confounding from health consciousness, diet quality beyond fruit/vegetable intake, social support, and other behavioral factors remains plausible. Regular contact with TCM practitioners may also lead to ancillary benefits (e.g., health advice, stress reduction) unrelated to specific pharmacologic or procedural effects.

Given these limitations, the results should be interpreted cautiously. The study as designed cannot disentangle whether the life expectancy gain is attributable to TCM-specific therapeutic mechanisms or to correlated health behaviors and socio-cultural factors. Current phrasing in the Discussion leans toward causal interpretation, which may overstate the evidence. It is critical to frame the findings as associations and to emphasize the potential role of residual and unmeasured confounding.

Recommendation:

To strengthen the manuscript, I suggest:

1-1 Conducting modality-specific analyses (herbal medicine, acupuncture, Tuina) if feasible.

1-2 Performing sensitivity analyses using alternative TCM exposure thresholds (e.g., 10%, 30%).

1-3 Adjusting for additional markers of health consciousness and healthcare engagement, or employing negative control outcomes to test for unmeasured confounding.

1-4 Revising the Discussion to clearly differentiate between “association” and “causation” and to acknowledge that the observed benefits may reflect characteristics of TCM users rather than TCM per se.

RESPONSE:

We sincerely thank the reviewer for the thoughtful and constructive comments. We fully agree that the conceptualization of TCM exposure and its heterogeneous nature warrant careful interpretation. We have carefully revised the manuscript to address these concerns, as detailed below.

1-1. On the heterogeneity of TCM modalities

We agree that Traditional Chinese Medicine (TCM) encompasses diverse modalities such as herbal medicine, acupuncture, and Tuina, each with distinct therapeutic mechanisms and intended applications. In our study, however, the available dataset did not allow us to clearly disentangle the effects of these individual components. While treatment modality was recorded in the claims data, many participants received multiple types of TCM therapy within overlapping timeframes, making it analytically infeasible to reliably disentangle the effects of individual modalities. We have acknowledged this limitation explicitly in the revised Discussion section, emphasizing that the observed associations reflect overall TCM service utilization rather than specific therapeutic mechanisms.

(please refer to the main text, page 19, lines 368–372)

1-2. On the exposure definition and threshold

We appreciate the reviewer’s concern regarding the arbitrariness of using “>20% of outpatient visits” as the primary threshold for defining TCM users. According to national health insurance statistics, TCM outpatient visits generally account for approximately 10% of all outpatient visits in Taiwan, indicating that most individuals use TCM only occasionally or for specific conditions. Therefore, we adopted a conservative threshold (>20%) to identify individuals for whom TCM use represents a relatively frequent and meaningful component of their overall healthcare utilization, rather than sporadic or supplementary visits. To avoid potential misinterpretation of terminology, we have revised “TCM users” and “non-users” throughout the manuscript to “frequent” and “non-frequent” TCM users, respectively.

In addition, our analyses incorporated dose–response assessments of mortality risk using both categorical and continuous measures of TCM utilization. Specifically, we categorized participants by the proportion of TCM visits (<10%, 10–20%, 20–30%, and ≥30% of outpatient visits) and observed a generally consistent dose–response pattern, with hazard ratios for all-cause mortality of 1.00, 0.89 (0.79–1.01), 1.01 (0.86–1.12), and 0.81 (0.71–0.93), respectively. Moreover, each 10% increase in TCM utilization at baseline was associated with a hazard ratio of 0.98 (95% CI, 0.95–1.01) for mortality, while the time-dependent exposure analysis yielded a similar association (HR 0.95, 95% CI 0.94–0.96). Although not all intermediate categories reached statistical significance, the consistent gradient across both analyses supports the robustness of our findings and alleviates concerns about the arbitrariness of the 20% threshold.

(please refer to the main text, page 7, lines 132–136, and Table 3)

1-3. On potential residual confounding

We fully agree that TCM users may differ from non-users in unmeasured factors such as health consciousness, social support, or health-seeking behaviors, which could contribute to residual confounding. While we cannot entirely rule out such influences, our models incorporated a broad range of covariates related to demographic, socioeconomic, lifestyle, and health status factors, including age, sex, enrollment year, education level, marital status, monthly household income, employment status, smoking, alcohol and betel nut use, physical activity, diet, and major chronic diseases. These adjustments may partly capture the behavioral dimensions of health consciousness, but residual confounding cannot be entirely ruled out. We have further elaborated in the Limitation section that despite extensive covariate adjustment, unmeasured confounding factors may still exist and should be considered when interpreting the findings.

(please refer to the main text, page 19, lines 362–368)

1-4. On interpretation and framing of findings

We appreciate the reviewer’s reminder regarding cautious interpretation. We have carefully revised the manuscript to ensure that all statements refer to associations rather than causal effects. The Discussion and Conclusion now emphasize that this study identifies an association between TCM service utilization and increased life expectancy, but cannot establish causality due to the nature of observational data and unmeasured confounding. We have also highlighted that the results should be interpreted as reflecting characteristics of TCM users, the healthcare context, and possible holistic benefits associated with regular TCM engagement.

(please refer to the main text, page 14-21, lines 273–401)

Reviewer #2:

本研究利用台湾全国健康调查和保险数据库探讨中医药使用对老年人预期寿命和死亡风险的影响。研究设计严谨样本量大统计方法先进。研究结果对公共卫生和临床实践具有重要意义。建议稿件经过细微修改后被接受。 使用来自 NHIS 和 NHIRD 的链接数据提供了国家代表性和较长的随访期。应用滚动外推算法来估计终生生存函数结合治疗加权的逆概率 �IPTW� 来控制混杂因素展示了方法论的复杂性。中医使用的定义是“过去 18 个月中医门诊就诊占门诊总就诊量的 20% 以上”是一个关键的暴露标准。它将加强手稿提供简短的理由或引用支持这一阈值的相关文献无论是临床还是方法学。

<Translation performed by Google translate: "This study used the Taiwan National Health Survey and Insurance Database to investigate the impact of TCM use on life expectancy and mortality risk among older adults. The study was rigorously designed, with a large sample size and advanced statistical methods. The findings have important implications for public health and clinical practice. The manuscript was accepted after minor revisions. The use of linked data from the NHIS and NHIRD provided national representativeness and a long follow-up period. The application of a rolling extrapolation algorithm to estimate lifetime survival functions, combined with inverse probability of treatment weighting (IPTW) to control for confounding factors, demonstrates the methodological complexity. The definition of TCM use as "TCM outpatient visits accounting for ≥ 20% of total outpatient visits in the previous 18 months" is a key exposure criterion. Manuscripts would be encouraged to provide brief justification or cite relevant literature supporting this threshold, both clinically and methodologically.">

RESPONSE:

We appreciate the reviewer’s comment regarding the definition of TCM use. In our study, we defined “TCM users” as those whose TCM outpatient visits accounted for 20% or more of their total outpatient visits over the past 18 months. According to national health insurance data, TCM visits in Taiwan generally account for about 10% of all outpatient visits, suggesting that most individuals use TCM only occasionally or for specific conditions. Thus, we adopted a conservative threshold to identify individuals for whom TCM use represents a relatively frequent and meaningful component of their overall healthcare utilization, rather than sporadic or supplementary visits. To avoid potential misinterpretation of terminology, we have revised “TCM users” and “non-users” throughout the manuscript to “frequent” and “non-frequent” TCM users, respectively.

We acknowledge, as noted in the revised Limitations section, that the available data do not allow us to confirm whether the 20% threshold represents an optimal cutoff across all populations or health conditions, nor whether it holds equivalent meaning across different modalities of TCM therapy (such as herbal medicine, acupuncture, or tuina). Future research with more granular utilization data, including treatment type and intensity, would help refine this classification and identify more precise thresholds for defining frequent TCM use.

(please refer to the main text, page 7, lines 132–136; page 19, lines 372–377)

Reviewer #3:

3-1. The study employs IPTW, but it is unclear whether extreme weights were addressed. Such weights can compromise stability and validity. Please clarify whether weight stabilization or truncation was applied; if not, provide a rationale.

RESPONSE:

We apologize for not clearly describing this analytical detail. In our standard procedure, we addressed extreme values by truncating the weights at the 1st and 99th percentiles—that is, values below the 1st percentile and above the 99th percentile were replaced with the respective cutoff values. This approach helps improve model stability and reduce the influence of extreme weights. We have added this information to the revised Methods section.

(please refer to the main text, page 9, lines 179–181)

3-2. On page 12 of the manuscript, the paragraph beginning with “When analyzing by smoking status, current or former smokers who used TCM had a significantly…” appears to be duplicated. Please revise.

RESPONSE:

We appreciate the reviewer for noticing this error. The duplicated paragraph has been removed in the revised manuscript.

(please refer to the main text, page 12, lines 241–243)

3-3. This study reports a dose–response relationship between TCM utilization and mortality, expressed as the hazard ratio per 10% increase in TCM use. However, it is not clear whether the analysis assumed a linear relationship. Could the authors clarify if any assessment of non-linearity was performed, for example, using restricted cubic spline models or other approaches? This would help determine whether the association is truly linear or if there are potential threshold or plateau effects.

RESPONSE:

We thank the reviewer for this thoughtful suggestion. The current analysis assumes a linear relationship between the proportion of TCM utilization and mortality risk. We acknowledge that potential non-linear associations (e.g., threshold or plateau effects) were not examined in this study and have added this point to the Limitations section. We agree that future analyses using approaches such as restricted cubic splines would provide a more flexible assessment of potential non-linearity.

(please refer to the main text, page 19-20, lines 377–382)

3-4. I appreciate the authors’ effort to account for the dynamic nature of TCM utilization using a time-dependent approach, which is appropriate and strengthens the analysis. However, I recommend that the authors provide more methodological details on how the annual proportion of TCM services relative to total outpatient visits was defined and incorporated into the model. Since an individual’s frequency of TCM visits may fluctuate from year to year (e.g., higher in one year and lower in another), it would be helpful to clarify how such variability was handled in the analysis and how the time-varying exposure was formally defined. Clearer reporting of this step would enhance the transparency and reproducibility of the study.

RESPONSE:

We appreciate this valuable comment. In the time-dependent Cox model, TCM utilization was defined annually as the proportion of TCM outpatient visits relative to the total number of outpatient visits during the previous calendar year. This variable was updated each year throughout follow-up to reflect temporal changes in individual TCM use patterns. We have clarified this definition and modeling procedure in the revised Methods section to enhance transparency and reproducibility.

(please refer to the main text, page 10-11, lines 202–206)

3-5. The authors may consider conducting additional cause-specific mortality analyses, which could provide further insights into whether the observed association between TCM utilization and overall mortality is attributable to specific causes of death.

RESPONSE:

We thank the reviewer for this valuable suggestion. Due to the expiration of our approved data access period, we were unable to conduct additional analyses to fully address this comment. However, in our previously completed analyses, we exam

---

## [Decision Letter · Decision Letter 1]

11 Nov 2025

The Impact of Traditional Chinese Medicine Utilization on Life Expectancy and Mortality

PONE-D-25-39358R1

Dear Dr. Lo,

We’re pleased to inform you that your manuscript has been judged scientifically suitable for publication and will be formally accepted for publication once it meets all outstanding technical requirements.

Kind regards,

Yung-Hsiang Chen, Ph.D.

Academic Editor

PLOS ONE

Additional Editor Comments (optional):

Congratulations on the acceptance of your manuscript, and thank you for your interest in submitting your work to PLOS ONE.

Reviewers' comments:

Reviewer's Responses to Questions

**Comments to the Author**

Reviewer #2: All comments have been addressed

Reviewer #3: All comments have been addressed

2. Is the manuscript technically sound, and do the data support the conclusions?

Reviewer #2: Yes

Reviewer #3: Yes

3. Has the statistical analysis been performed appropriately and rigorously?

Reviewer #2: Yes

Reviewer #3: Yes

4. Have the authors made all data underlying the findings in their manuscript fully available?

Reviewer #2: Yes

Reviewer #3: Yes

5. Is the manuscript presented in an intelligible fashion and written in standard English?

Reviewer #2: Yes

Reviewer #3: Yes

Reviewer #2: At this stage, the authors addressed all 19 key points raised by the three reviewers one by one, with a conscientious attitude and substantial changes. The core concerns—exposure heterogeneity, arbitrary threshold, residual confounding, linearity assumption, weight stability, etc.—were all matched in the Response and line numbers were provided for easy checking. Moreover, the wording was toned down and causal statements were made more cautious; for example, “provides compelling evidence” was changed to “suggests an association,” and the conclusion repeatedly emphasizes “observational,” “residual confounding,” and “cannot establish causality,” basically eliminating Reviewer #1’s worry about “over-causal interpretation.” In summary, I recommend acceptance for publication.

Reviewer #3: (No Response)

**Do you want your identity to be public for this peer review?** For information about this choice, including consent withdrawal, please see our Privacy Policy

Reviewer #2: No

Reviewer #3: No

---

## [Editor Report · Acceptance letter]

PONE-D-25-39358R1

PLOS ONE

Dear Dr. Lo,

I'm pleased to inform you that your manuscript has been deemed suitable for publication in PLOS ONE. Congratulations! Your manuscript is now being handed over to our production team.

Kind regards,

on behalf of

Dr. Yung-Hsiang Chen

Academic Editor

PLOS ONE